# Measurements of $t\bar{t}$ in association with charm quarks at 13 TeV with the ATLAS experiment

**Knut Zoch[1⋆], on behalf of the ATLAS Collaboration**

**1** Laboratory for Particle Physics and Cosmology, Harvard University,
Cambridge, Massachusetts 02138, USA

⋆ kzoch@g.harvard.edu

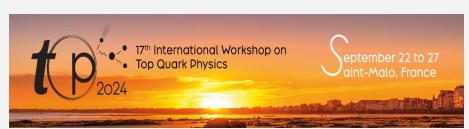

*The 17th International Workshop on
Top Quark Physics (TOP2024)
Saint-Malo, France, 22-27 September 2024*

## Abstract

**This talk presents the ATLAS Collaboration's first measurement of the inclusive cross-section for top-quark pair production in association with charm quarks. Using the full Run 2 proton–proton collision data sample at $\sqrt{s} = 13$ TeV, collected with the ATLAS experiment at the LHC between 2015 and 2018, the measurement selects $t\bar{t}$ events with one or two charged leptons and at least one additional jet in the final state. A custom flavour-tagging algorithm is employed to simultaneously identify $b$-jets and $c$-jets. The fiducial cross-sections for $t\bar{t}+\geq2c$ and $t\bar{t}+1c$ production are found to largely agree with predictions from various $t\bar{t}$ simulations, though all underpredict the observed values.**

## 1 Introduction

Top-quark pair production ($t\bar{t}$) with additional heavy-flavour jets is a large irreducible background to many other rare processes predicted by the Standard Model (SM) of particle physics. Prominent examples include $t\bar{t}H$ production with $H \rightarrow b\bar{b}$ decays and the production of four top quarks ($t\bar{t}t\bar{t}$) in single-lepton or dilepton final states. In this context, *heavy flavour* refers to jets originating from $b$-quarks or $c$-quarks. In $t\bar{t}$ events, additional heavy-flavour jets can arise from gluon splitting into $b\bar{b}$ and $c\bar{c}$ pairs. The $b\bar{b}$ or $c\bar{c}$ pair can form separate jets ($t\bar{t} + b\bar{b}/c\bar{c}$) or be clustered into a single jet ($t\bar{t} + 1B/C$). In addition, a single additional $b$-quark or $c$-quark can originate from the initial state ($t\bar{t} + 1b/t\bar{t} + 1c$). Illustrative Feynman diagrams for these processes are shown in Figure 1. Computations of $t\bar{t} + b\bar{b}$ production cross-sections exist at next-to-leading order (NLO) accuracy in quantum chromodynamics (QCD), but uncertainties in the choice of the renormalisation and factorisation scales remain sizeable due to the different energy scales involved in the process. Currently, no dedicated $t\bar{t} + c\bar{c}$ computations are available, emphasising the need for experimental measurements to improve the understanding of these final states.

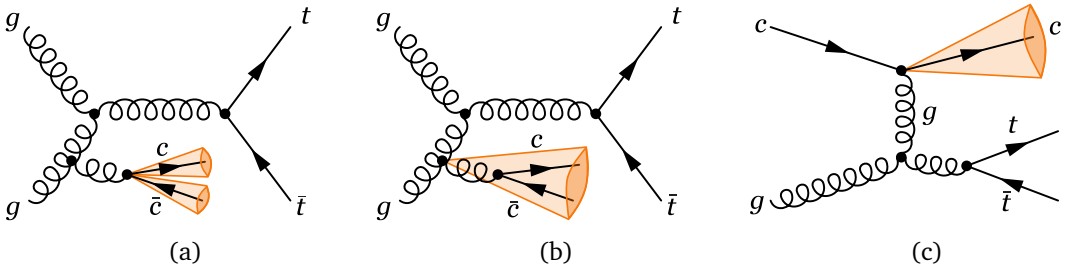

Figure 1: Illustrative Feynman diagrams for $t\bar{t}$ pair production with one or multiple additional $c$-jets: (a) $t\bar{t} + c\bar{c}$ production via initial-state gluon radiation where both $c$-quarks form a jet each; (b) $t\bar{t} + c\bar{c}$ production via initial-state gluon radiation where the two $c$-quarks are in the same jet; (c) $t\bar{t} + 1c$ production where the $c$-quark originates from the initial state. Figure taken from Ref. [3].

The ATLAS Collaboration [1] at the Large Hadron Collider (LHC) [2] at CERN has now performed the first measurement of the inclusive cross-section for $t\bar{t}$ production in association with charm quarks [3] to improve the understanding of these final states. This measurement uses the full Run 2 proton–proton ATLAS dataset at $\sqrt{s} = 13$ TeV taken between 2015 and 2018, with an integrated luminosity of $140$ fb$^{-1}$. The measurement selects $t\bar{t}$ events with one or two charged leptons and at least one additional jet in the final state. A custom flavour-tagging algorithm is employed to simultaneously identify $b$-jets and $c$-jets. For the first time, this measurement then uses regions sensitive to the presence of $c$-quarks to extract separate fiducial cross-sections for $t\bar{t} + {\geq}2c$ and $t\bar{t} + 1c$ production. The first includes events with at least two $c$-jets, targeting primarily the $t\bar{t} + c\bar{c}$ process illustrated in Figure 1a, while the latter comprises events with one additional $c$-jet, originating from either the $t\bar{t} + 1C$ scenario (Figure 1b), from $t\bar{t} + 1c$ production (Figure 1c), or from $t\bar{t} + c\bar{c}$ where one of the $c$-jets falls outside the acceptance phase space of the detector. The CMS Collaboration previously measured $t\bar{t} + c\bar{c}$ cross-sections consistent with $t\bar{t}$ simulation predictions [4].

This document and the associated talk summarise the methodology and results of the new ATLAS measurement [3]. Detailed references to all methods, calculations, software tools, etc. are skipped here for brevity but can be found in the original publication.

## 2 Simulation of signal and background

One key component in scrutinising $t\bar{t}$ production in association with heavy-flavour jets is the modelling of these processes through Monte Carlo (MC) simulations. Recent ATLAS measurements of $t\bar{t} + b\bar{b}$ and $t\bar{t}H(H \to b\bar{b})$ production [5,6] have found the modelling of $t\bar{t} + {\geq}1c$ to be a limiting factor in the precision of the measurements. In the absence of dedicated $t\bar{t} + c\bar{c}$ computations, inclusive $t\bar{t}$+jets simulations at NLO accuracy interfaced to a parton shower (PS) (NLO+PS) provide the best available model for $t\bar{t} + {\geq}1c$. Here, the gluon splittings into $c\bar{c}$ pairs are modelled through the PS algorithm, not in the matrix-element calculation. $t\bar{t} + {\geq}1b$ production, on the other hand, can be modelled through $t\bar{t} + b\bar{b}$ four-flavor scheme (4FS) matrix elements, where the $b$-quark is treated as a massive particle and the production of the $b\bar{b}$ pair is described directly in the calculation of the hard interaction.

In the ATLAS measurement, $t\bar{t}$+jets simulations in the five-flavor scheme (5FS), generated with POWHEG BOX 2 [7–10] and the NNPDF3.0NLO 5FS parton distribution function (PDF) set [11], are used to model the $t\bar{t} + {\geq}1c$ and $t\bar{t}$+light processes. $t\bar{t} + b\bar{b}$ 4FS simulations using POWHEG BOX RES [12], OPENLOOPS [13–15] and the NNPDF3.0NLO 4FS PDF set [11] are em-

ployed for the $t\bar{t} + \geq 1b$ process. Both are interfaced to PYTHIA 8 [16] for the simulation of the PS and hadronisation using the ATLAS A14 set of tuned parameters [17] and the NNPDF2.3LO PDF set [18]. Reweightings and alternative setups (POWHEG BOX 2+HERWIG 7 [19–21], varied NLO matching) assess modeling uncertainties. For comparison only, an additional set of $t\bar{t}$+jets events is generated using MADGRAPH5_AMC@NLO [22] interfaced to HERWIG 7.

All simulated events are processed through the full ATLAS detector simulation and reconstruction chain, and are then split into four categories: $t\bar{t} + \geq 2c$, $t\bar{t} + 1c$, $t\bar{t} + \geq 2b$, and $t\bar{t} +$ light. The categorisation is based on the presence of additional $b$-jets and $c$-jets clustered at the stable particle level before detector simulation. Those events with one or more $b$-jets are assigned to the $t\bar{t} + \geq 2b$ category, those with one $c$-jet to the $t\bar{t} + 1c$ category, and those with at least two $c$-jets to the $t\bar{t} + \geq 2c$ category. The latter two classes of events must not contain any $b$-jets. All remaining events are assigned to the $t\bar{t} +$ light category. The categorisation is used to combine the $t\bar{t}$+jets and $t\bar{t} + b\bar{b}$ simulations by removing all $t\bar{t} + \geq 1b$ events from the 5FS $t\bar{t}$+jets sample and all but the $t\bar{t} + \geq 1b$ events from the 4FS $t\bar{t} + b\bar{b}$ sample, avoiding double-counting.

Various other processes are considered as background to the measurement, including single-top production, associated production of single top quarks or top-quark pairs with a vector boson, and $W +$ jets, $Z +$ jets and diboson production. Dedicated MC simulations are used to model these processes. Background contributions from events with fake-lepton signatures are estimated using data-driven methods in the single-lepton channel and from MC simulations in the dilepton channel. Uncertainties in the dominant background processes are assessed through dedicated modelling uncertainties, while conservative normalisation uncertainties are assigned to the minor backgrounds.

## 3 Analysis strategy

A dedicated flavour-tagging algorithm, the $b/c$-tagger, was developed for this analysis to simultaneously identify $b$-jets and $c$-jets with high efficiency. While the standard ATLAS DL1r $b$-tagging algorithm [23] provides high efficiency for $b$-jets, no $c$-tagging working points are available, mandating this custom approach. The output scores provided by DL1r were re-optimised into a two-dimensional discriminant defining five $b/c$-tagger bins: two each for $c$-jet ($c$@11%, $c$@22%) and $b$-jet ($b$@60%, $b$@70%) tagging, and one for untagged jets, as illustrated in Figure 2. When using the looser tagging bins, the tighter bins are implicitly included, e.g., the $c$@22% bin includes all jets that pass $c$@11%.

The analysis uses a suite of single-electron and single-muon triggers to select events with one or two charged leptons ($e$ or $\mu$), of which one must be matched to the trigger object. The single-lepton channel selects one lepton and $\geq 5$ jets, while the dilepton channel selects two leptons and $\geq 3$ jets, with at least three jets passing $b$@70% or $c$@22% criteria. Same-flavour lepton pairs are vetoed for low invariant masses and around the $Z$ boson mass. The single-lepton channel is split into events with five jets and events with six or more jets (5-jet-exclusive and 6-jet-inclusive). The dilepton channel is split into 3-jet-exclusive and 4-jet-inclusive events.

The five $b/c$-tagger bins define 19 orthogonal control regions (CRs) and signal regions (SRs) across the single-lepton and dilepton channels, providing sensitivity to $t\bar{t} + \geq 2c$, $t\bar{t} + 1c$, and controlling background contributions. The twelve CRs are designed to either exclude $c$-tagged jets or select events with exactly one $c$-tagged jet, isolating $t\bar{t} + \geq 1b$, $t\bar{t} +$ light, and other background contributions. The seven SRs require at least two (one) jet tagged with $c$@22% in the single-lepton (dilepton) channel and have different predicted purities of $t\bar{t} + \geq 2c$ and $t\bar{t} + 1c$ events. These regions are then used to extract the $t\bar{t} + \geq 2c$ and $t\bar{t} + 1c$

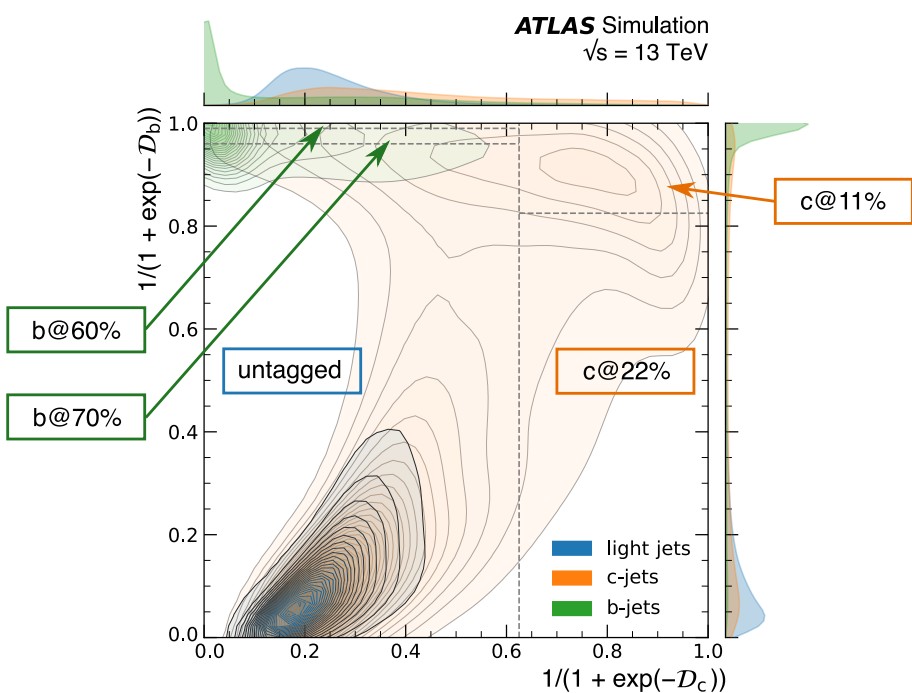

Figure 2: Distribution of light, $c$-, and $b$-jets for the two-dimensional $b/c$-tagger in simulated $t\bar{t}$ events. Dashed lines correspond to the edges of the five $b/c$-tagger bins. For visualisation purposes a standard logistic function is applied to both axes of the discriminant. Figure taken from Ref. [3].

signal strengths from the data using a profile likelihood fit. Additionally, normalisation factors for the $t\bar{t} + {\geq}1b$ and $t\bar{t} +$ light background contributions are included in the fit and are left free-floating. Systematic uncertainties relating to the modelling of the signal and background processes, the dedicated flavour-tagging algorithm, and all other instrumental uncertainties are included in the fit as nuisance parameters constrained by Gaussian penalty terms. While the CRs are included as a single bin in the fit, the jet-inclusive and jet-exclusive SRs use jet-multiplicity and invariant-mass distributions of the $c$-tagged jets, respectively.

The measurements are performed in a fiducial phase defined at the stable particle level that mimics the selection criteria of the analysis. Apart from the lepton requirements, the fiducial phase requires at least one additional jet, but no explicit requirements are imposed on its flavour. The measurements are also performed in a more inclusive volume without requirements on the $t\bar{t}$ decay products and the jet multiplicity.

## 4   Results

Good post-fit agreement between data and simulation is observed in all CRs and SRs of the analysis. The goodness of fit was evaluated using a *saturated model* and the compatibility with data was found to be 98 %. Using the extracted $t\bar{t} + {\geq}2c$ and $t\bar{t} + 1c$ signal strengths, the fiducial cross-sections are determined to be

$$\sigma^{\text{fid}}(t\bar{t} + {\geq}2c) = 1.28\,^{+0.16}_{-0.10}\,(\text{stat})\,^{+0.21}_{-0.22}\,(\text{syst})\,\text{pb} = 1.28\,^{+0.27}_{-0.24}\,\text{pb}, \tag{1}$$

$$\sigma^{\text{fid}}(t\bar{t} + 1c) = 6.4\,^{+0.5}_{-0.4}\,(\text{stat})\,\pm 0.8\,(\text{syst})\,\text{pb} = 6.4\,^{+1.0}_{-0.9}\,\text{pb}. \tag{2}$$

The largest contributing sources of uncertainties are the modelling of $t\bar{t} + {\geq}1c$, $t\bar{t} + {\geq}1b$, and $t\bar{t} +$ light, in particular in the NLO matching and the PS, the uncertainties in the $b/c$-tagger,

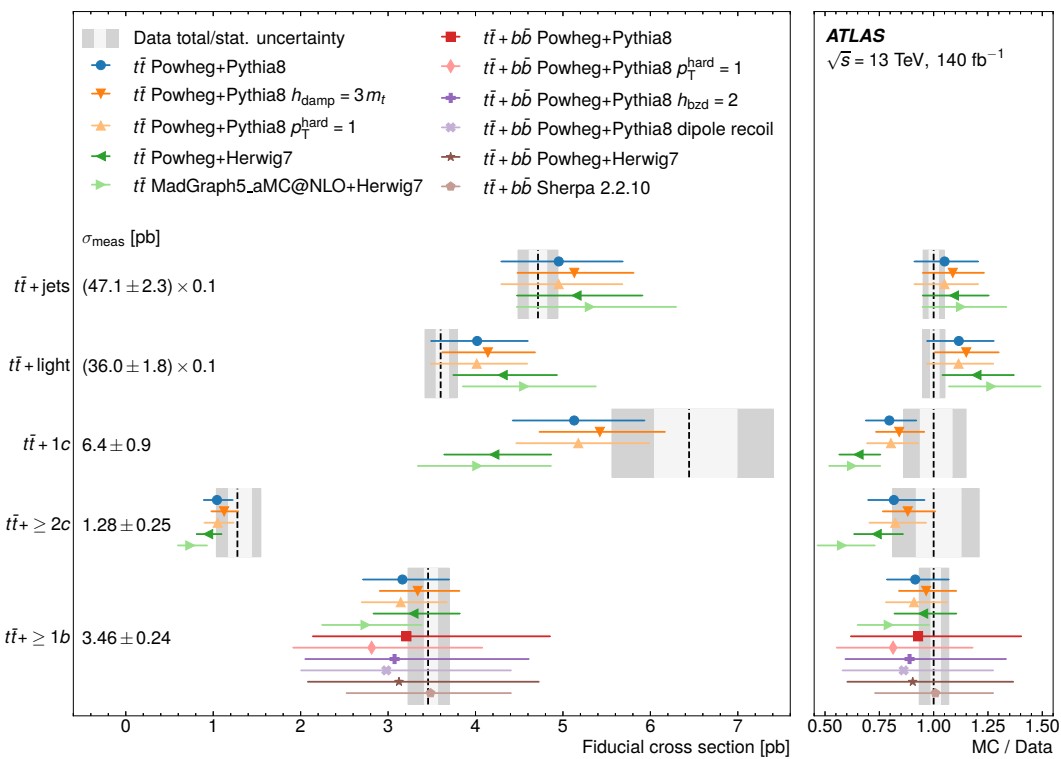

Figure 3: Measured fiducial cross-section values in comparison with various NLO+PS predictions from inclusive $t\bar{t}$+jets and $t\bar{t}+b\bar{b}$ simulations. Figure taken from Ref. [3].

and the data statistics. The measured $t\bar{t} + {\geq}1b$ normalisation factor is compatible with those obtained in the dedicated ATLAS $t\bar{t} + b\bar{b}$ measurements [5].

The measured values are compared to various NLO+PS predictions from inclusive $t\bar{t}$+jets and $t\bar{t}+b\bar{b}$ simulations in Figure 3. The prediction for $t\bar{t}+{\geq}2c$ and $t\bar{t}+1c$ are largely consistent with the measured values, though all predictions underpredict the observed values. Considering scale and PDF variations on the predictions, the $t\bar{t}$ POWHEG BOX 2+PYTHIA 8 predictions agree within 1.1 standard deviations of measurement and prediction uncertainties. Agreement between 1.2 and 2.0 standard deviations is observed for the POWHEG BOX 2+HERWIG 7 and the MADGRAPH5_AMC@NLO+HERWIG 7 predictions. Fit stability tests, including combined and independent $t\bar{t} + {\geq}2c$ and $t\bar{t} + 1c$ parameterisations, confirm robustness.

In addition to the fiducial cross-sections, the ratios of $t\bar{t} + {\geq}2c$ and $t\bar{t} + 1c$ to total $t\bar{t}$+jets production are extracted in the fiducial and the more inclusive volume. The values were found to be $R^{\mathrm{inc}}_{t\bar{t}+{\geq}2c} = (1.23 \pm 0.25)\%$ and $R^{\mathrm{inc}}_{t\bar{t}+1c} = (8.8 \pm 1.3)\%$ in inclusive volume, in agreement with NLO+PS simulations at a similar level as the fiducial cross-sections.

## 5 Conclusion

The ATLAS Collaboration at the LHC has performed their first measurement of the inclusive cross-section for $t\bar{t}$ production in association with charm quarks [3]. The measurement uses the full Run 2 proton–proton collision data sample at $\sqrt{s} = 13\,\mathrm{TeV}$, corresponding to an integrated luminosity of $140\,\mathrm{fb}^{-1}$, collected between 2015 and 2018. Using single-lepton and dilepton events and a custom flavour-tagging algorithm to identify $b$-jets and $c$-jets, the fiducial cross-sections for $t\bar{t} + {\geq}2c$ and $t\bar{t} + 1c$ production are extracted separately from the data

for the first time. The measured values are found to be in agreement with various NLO+PS predictions from inclusive $t\bar{t}$+jets and $t\bar{t} + b\bar{b}$ simulations, though all predictions underpredict the observed values. Precision is limited by modelling uncertainties ($t\bar{t} + {\geq}1c$, $t\bar{t} + {\geq}1b$, $t\bar{t} +$ light), uncertainties in the $b/c$-tagger, and data statistics. Measurements of the ratios of the $t\bar{t} + {\geq}2c$ and $t\bar{t} + 1c$ cross-sections to the inclusive $t\bar{t}$+jets cross-section are also performed and found to be in agreement with the predictions from NLO+PS simulations at a similar level as the fiducial cross-sections.

## Acknowledgements

The author received support from the U.S. Department of Energy Award No. DE-SC0007881.

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
