# Peer review of "Measurements of $t\bar{t}$ in association with charm quarks at 13 TeV with the ATLAS experiment"

_SciPost Physics Proceedings_

## Round 1 · Referee Report · Anonymous (Referee 1) · 2025-5-20

Report

This document reports on the first ATLAS measurement of inclusive cross sections for top quark pair (tt) production in association with charm quarks. The measurements are performed using the full set of data collected by the ATLAS experiment at 13 TeV during the LHC Run2 and target final states with one or two charged leptons and at least one additional jet. A dedicated heavy-flavour tagging algorithm is developed to identify jets originating from b- and c-quarks (b- and c-jets, respectively). For the first time, inclusive cross sections are measured separately for tt plus 1c-jet and tt plus at least 2 c-jets, and are compared to tt Monte Carlo simulations at Next-to-Leading-Order accuracy in QCD. These measurements provide crucial inputs for improving the modelling of tt production with additional heavy flavour jets, which is essential for precision measurements of rare Standard Model processes and searches for new physics phenomena, where tt+c-jets is an important background.

The analysis is solid and well described in the document, providing relevant information to the audience of SciPost. Therefore I recommend its publication in your journal once the authors have addressed the (minor) comments posed.

Requested changes

  • Section 1

  • 1st parag: —“b-quarks or c-quarks” —> “b-quarks or c-quarks (referred to as b-jets and c-jets, respectively)” or similar — “[…] clustered into a single jet […]” —> what happens in the case where 1b/c is out of acceptance?

  • 2nd parag: — 1st sentence: the phrasing is misleading. It implies that the ATLAS Collaboration has performed the first measurement of associated tt production with additional charm quarks, while the first ever measurement of this process was done by CMS. Suggest the following rephrasing: “The ATLAS Collaboration [1] at the CERN Large Hadron Collider (LHC) [2] has recently presented a measurement of the inclusive cross-section for tt production in association with charm quarks [3], aimed at improving the understanding of these final states. "

— “with an integrated” —> “corresponding to an integrated” — “two charged leptons” —> “two charged leptons (e or mu)” — “For the first time […]” —> “For the first time, regions sensitive to the […] are used in this measurement to extract separate […]” — “tt+>=2c” is undefined — “The first” —> “The former” — “The CMS Collaboration […]” —> “The CMS Collaboration performed the first measurement of tt+cc cross-sections, finding consistent results with SM tt predictions at NLO accuracy in QCD [4]”

  • Section 2

  • 1st parag: “tt+>=1b production” —> “The tt+>=1b process”

  • 2nd parag: — “tt+light” undefined — “[…] assess modeling […]” —> “[…] are employed to assess modeling […]”
  • 3rd parag: “are then split” —> “are further split”
  • 4th parag: — “Various other” —> suggest to remove “other” — “single top” —> “single top quark” — “fake-lepton” is undefined — “Uncertainties […] minor backgrounds” —> it is not clear which background processes are dominant or minor. Maybe worth adding a sentence. — “conservative” —> Suggest to remove, as it does not mean anything without further explanation.

  • Section 3

  • 1st parag: “[…] mandating this custom approach” —> Suggest: “ […] the lack of available c-tagging working points necessitates this custom approach”

  • 2nd parag:
    — “two charged leptons (e or mu)” —> “two charged e or mu” — “trigger object” is jargon, but ok — “with at least three jets […]” —> does this requirement apply to both single-lepton and dilepton channels? It is not clear.
  • 3rd paragraph: — “signal strength” is undefined — “[…] and left free-floating” —> “[…] and treated as free-floating parameters.”
  • 4th paragraph: — “phase” —> “phase space” (appearing twice)

  • Section 4 — “Fit stability tests […] confirm robustness” —> “Stability tests […] validate the robustness of the fit” — “[…] the more inclusive volume” —> it is not clear what “more inclusive” exactly refers to

  • Section 5 — “the fiducial cross-sections” —> “fiducial cross-sections”, as the phase space is not defined here —“measured values” —> “measurements” — “data statistics” —> “size of the data sample”

  • Bibliography:

  • Ref. [3] and all others where it applies: the link to the arXiv version seems to be broken, please check

Recommendation

Publish (surpasses expectations and criteria for this Journal; among top 10%)

---

## Editorial Decision

resubmitted